# Template-in-template assembly nanostructured microspheres for high performance chromatography

Juxing Zeng[1,2,5], Hanchen Cao[1,2,5], Kaiyue Sun[1,2,5], Zhuoheng Zhou[3,5], Lin Lv[1,2], Jikai Chen[1,2], Xiangyu Huang[1,2], Xiaofei Wang[4] & Bo Zhang ®[1,2] ✉

Concurrent precision architecture of morphology and nanostructure in mesoporous microspheres is pivotal for high performance separations. Herein, we develop a template-in-template assembly nanostructuring (TiTAN) strategy to precisely synthesize monodisperse microspheres with ordered mesoporous nanostructure. Microfluidic droplet templating ensures uniform particle morphology (CV = 3%), while structure-directing agents within droplets enable tailored pore configurations, including 2D hexagonal, body-centered cubic, face-centered cubic, and cubic double gyroidal mesostructures. Through regulating hydrothermal conditions, structural parameters can be fined-tuned with 2 Å spatial resolution. By extending this manufacturing capability to a variety of material chemistries, chromatographic materials can now be de novo architectured with rationales, with the performance demonstrated by the solution of a classical separation challenge: resolving critical pairs (whose selectivity, α, infinitely approaching to 1), and with the shortest possible time. Beyond separation medium, the TiTAN strategy also suggests a route towards general synthesis of porous material with precision macroscopic morphology and microscopic nanostructure.

Mesoporous materials[1,2], known for their suitable pore size for mass transfer and large surface area for molecular adsorption/partitioning, have emerged as an important separation medium and been widely used in the fields of chemical separation[3], environmental analysis[4], and pharmaceutical purification[5]. Given the essential demands of separation performance in terms of resolution, efficiency and speed, it is of great significance to precise design the structure of the separation medium, which involves the simultaneous control of both the external morphology and internal mesoporous nanostructure of microspheres. The morphology of microspheres (including shape, size, and size distribution) determines the laminar flow field, which affects eddy dispersion of the fluidic transportation process[6–8]. On the other hand, the mesoporous structure, including

pore size, surface area, pore volume and pore configuration, is highly relevant to the intraparticle mass transfer process, which collectively contributes to the separation kinetics[9–11]. Therefore, the development of a robust synthesis strategy enabling precise control over both the microsphere's morphology and mesopore's structure is of paramount significance for high-performance chromatography.

Current solution-phase synthetic methodologies can afford fine control of the morphology and dispersity of mesoporous nanoparticles[12], however, pushing this level of controllability up to micrometer-scale remains challenging[13]. During the formation of microspheres, the heterogeneous nucleation process and growth kinetics, as well as insufficient colloidal stabilization, result in a broad size distribution and non-uniform morphology[14–16]. To address these

[1]Department of Chemistry and the MOE Key Laboratory of Spectrochemical Analysis & Instrumentation, College of Chemistry and Chemical Engineering, Xiamen University, Xiamen, China. [2]State Key Laboratory of Vaccines for Infectious Diseases, Xiang An Biomedicine Laboratory, Xiamen University, Xiamen, China. [3]Bioanalytical Services, Department of Toxicology, WuXi AppTec (Suzhou) Co. Ltd, Suzhou, China. [4]ColumnScientific Inc., Xiamen, China. [5]These authors contributed equally: Juxing Zeng, Hanchen Cao, Kaiyue Sun, Zhuoheng Zhou. ✉e-mail: bozhang@xmu.edu.cn

issues, strategies such as polymer-seed swelling, core-shell self-assembly, hard templating, and emulsion-based synthesis have been developed to produce mesoporous microspheres with tailored shape and narrow dispersity[17–21]. Nevertheless, the pore networks in these materials typically emerge from stochastic processes–phase separation or random condensation of precursors–which fundamentally restricts precise control of mesoporous nanostructure, thereby compromising chromatographic performance and applicability[19,20].

Over the years, various strategies have been introduced to precise control the orderliness of pore structure of microspheres[22–27]. The development of ordered mesoporous silica (e.g., SBA-15 and MCM-41) has realized uniform nanostructure and long-range periodicity through cooperative assembly of surfactant templates and guest species, or by iterative nanocasting of mesoporous solids[28–31]. Crystalline frameworks, i.e., metal-organic frameworks (MOFs) and covalent organic frameworks (COFs), exploit reversible bonding (metal-ligand coordination in MOFs; dynamic covalent bonds in COFs) to facilitate error correction and network rearrangement, thereby affording thermodynamically stable, highly ordered pore lattices[32–34]. Nevertheless, the formation of these ordered nanoporous materials is based on the orientational growth of the nanostructure units, which inevitably results in irregular or crystalline morphologies rather than "real spheres". The difficulty, if not completely infeasible, of assembling such non-spherical materials into uniform and permeable beds, as required to achieve satisfactory separation kinetics, limits their chromatography applicability. Consequently, the development of methodologies which can enable precise tuning of ordered mesoporous structures while maintaining spherical and uniform morphology, remains an unmet synthetic challenge.

Herein, we report a template-in-template assembly nanostructuring (TiTAN) strategy based on a microfluidic platform, in order to realize precision synthesis of microspheres with uniform spherical morphology and well-defined mesoporous structure. The sphere morphology and monodispersity are precisely modulated by microdroplet templating, while the ordered mesopores inside are independently structured via colloidal templating. By extending this manufacturing capability to a variety of spherical sizes, pore configurations and material chemistries, chromatographic media can now be de novo architectured with rationales, with separation performance realized by design. These advantages have been demonstrated by the solution of an old, but long-pursued separation challenge: resolving critical pairs (whose selectivity factor, $\alpha$, infinitely approaching to unit), and with the shortest possible time.

## Results

Monodisperse microspheres with ordered mesoporous structure were synthesized via a TiTAN strategy based on droplet microfluidic technology (Fig. 1a). Specifically, the dispersed phase consisted of an acidic volatile organic solvent containing amphiphilic surfactant (e.g., Pluronic PEO-b-PPO-b-PEO) as structure-directing agent (SDA) and hydrolysate. Using the previously developed microfluidic platform[35], this phase was emulsified into uniform micro-sized droplets with fluorinated oil (HFE-7500) containing fluorosurfactants (e.g., perfluoropolyether-b-polyethylene, glycol-b-perfluoropolyether) as the continuous phase (Supplementary Fig. 1). Following collection in FC-

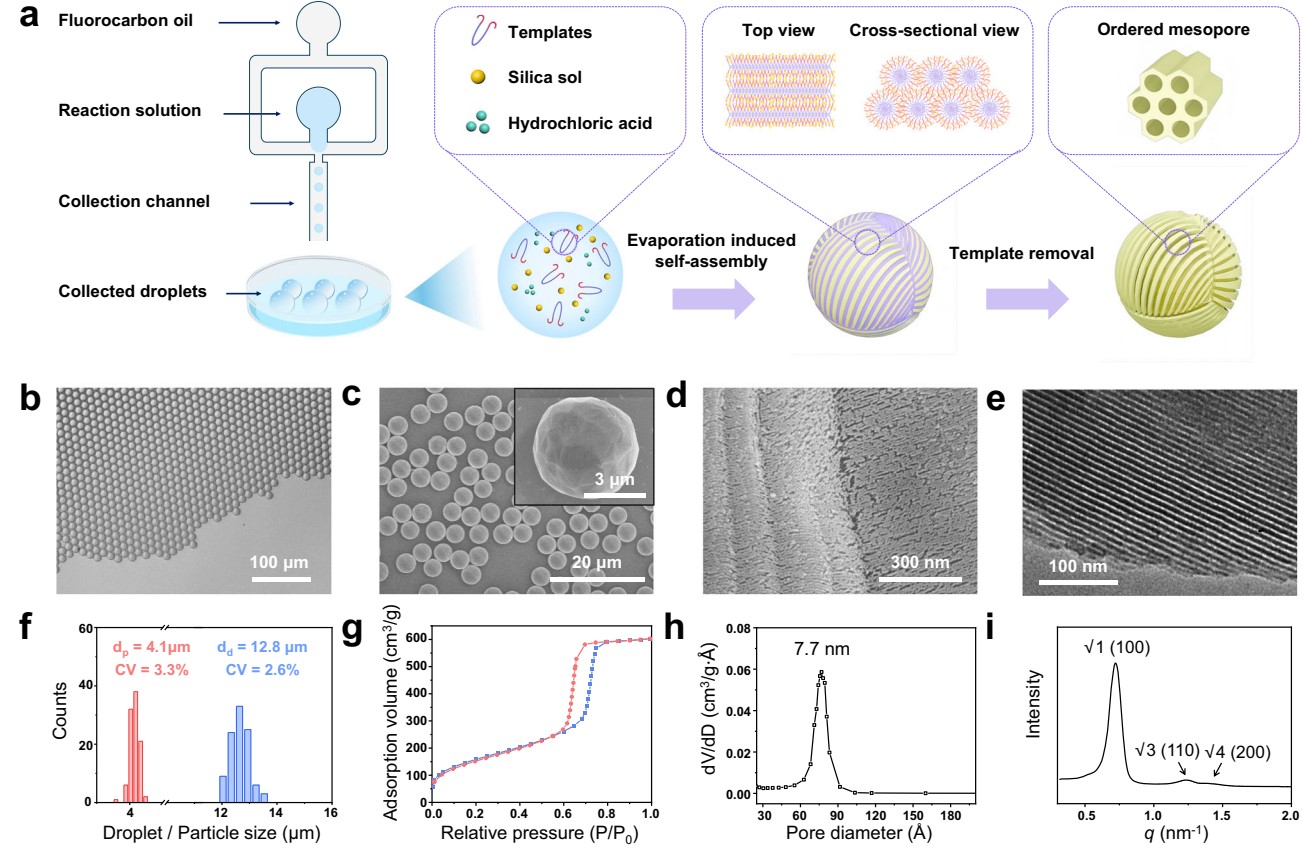

**Fig. 1 | Synthesis and characterization of monodisperse microspheres with ordered mesopore structure. a** Schematic of the synthesis of ordered mesoporous silica microspheres. **b** Optical microscopic image of monodisperse droplets generated by the microfluidic device. **c, d** SEM images with different magnifications and **e** TEM image of the obtained ordered mesoporous silica microspheres. **f** Size distribution analysis of the monodisperse droplets ($d_d$, blue) and microspheres ($d_p$, red). **g** Nitrogen adsorption-desorption isotherm, **h** pore size distribution, and **i** SAXS pattern of the ordered mesoporous silica microspheres.

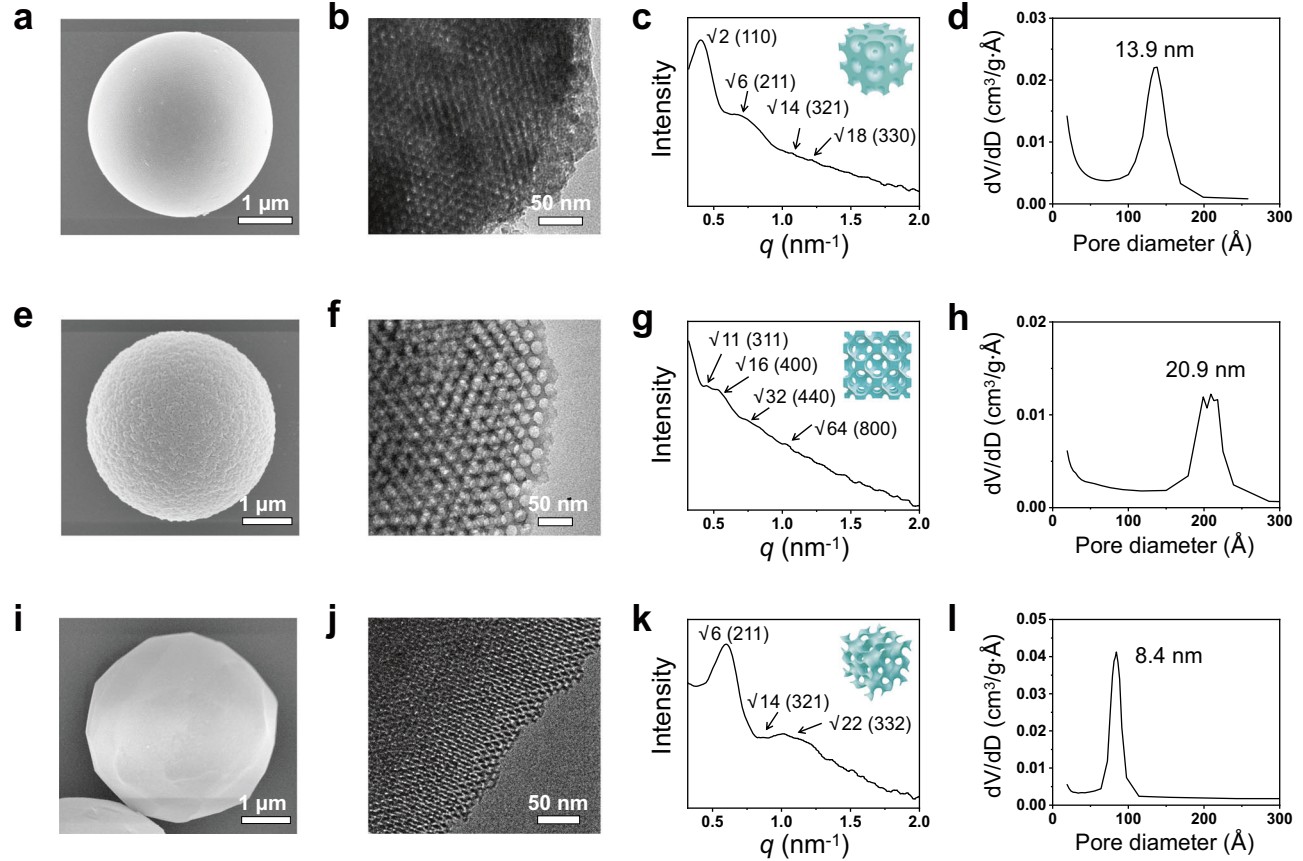

**Fig. 2 | Structural characterization of the monodisperse microspheres with different pore configurations.** SEM images, TEM images, SAXS patterns, and pore size distributions of **a**–**d** body centered cubic, **e**–**h** face centered cubic, and **i**–**l** cubic double gyroidal configurations, respectively.

40 carrier oil ($\rho = 1.85\,g/cm^3$), the generated droplets floated to the liquid-air interface, enabling fast solvent evaporation (Supplementary Fig. 2 and Supplementary Movie 1). During evaporation, the progressive concentration of amphiphilic surfactants and precursors induced the formation of liquid-crystalline phase, while simultaneously promoting the condensation of surrounding silica species. In this process, the droplets served both as microscale reaction vessels and morphology templates for microsphere formation. Ultimately, monodisperse microspheres with well-defined mesoporosity and regular spherical morphology were prepared, followed by subsequent hydrothermal treatment and calcination to remove organic residues and enhance framework crosslinking.

As an example, silica microspheres with 2D hexagonal (2D-hex) mesostructure were prepared using P123 triblock copolymer as SDA and tetraethoxysilane (TEOS) as reaction precursor. As a result of emulsification in microchannels, monodisperse droplets were generated with a diameter of 12.8 μm (CV = 2.6%, Fig. 1b, f), and the corresponding silica microspheres were obtained following solidification after solvent evaporation. Large-scale field emission scanning electron microscopy (FE-SEM) revealed that the silica microspheres have an average diameter of 3.9 μm with a narrow size distribution (CV = 2.4%, Supplementary Fig. 3). Following hydrothermal treatment at 130 °C for 24 h, the microsphere size slightly increased to 4.2 μm, and finally decreased to 4.1 μm after calcination at 550 °C for 6 h. The microsphere size distribution remained monodisperse throughout the post-treatment process (CV = 2.8% after hydrothermal treatment and 3.3% after calcination, Fig. 1f and Supplementary Fig. 3). Instead of appearing as a smooth spherical surface, the calcined silica microspheres exhibited as sphere-like polyhedrons composed of multiple planes, which were found to consist of parallel arrangements of

cylindrical silica rods (Fig. 1c, d). Characterized by transmission electron microscopy (TEM), highly ordered mesoporous frameworks with pore size of approximately 8 nm were observed along the [110] facet (Fig. 1e). Small-angle X-ray scattering (SAXS) analysis showed that the silica microspheres have three distinct and sharp diffraction peaks indexed to the 100, 110, and 200 reflections, respectively, demonstrating the formation of an ordered 2D-hex mesoporous configuration (*p6mm* space group, Fig. 1i). The nitrogen adsorption-desorption isotherm of the silica microspheres exhibited a representative type IV curve with a sharp hysteresis loop at $P/P_O = 0.6 - 0.7$ (Fig. 1g). The pore size has a narrow distribution at 7.7 nm, further confirmed the well-defined mesoporosity of the material (Fig. 1h).

While various types of ordered mesoporous structured materials have been designed and synthesized, hexagonal and cubic symmetries represent the most prominent and well-studied classes. By adjusting the SDAs and post-treatment conditions, the TiTAN strategy enables the precision construction of these typical mesostructures in monodisperse spherical materials (Fig. 2a, e, i, and Supplementary Figs. 5–7).

Common mesostructures with cubic symmetry include body centered cubic (bcc), face centered cubic (fcc) and cubic double gyroidal (cdg) structures. Bcc configuration can be formed by SDAs with a high hydrophilic/hydrophobic ($V_H/V_L$) volume ratio as templates (Fig. 2b). The silica microspheres with a $Im\bar{3}m$ cubic configuration were synthesized by using F127 as SDA ($Im\bar{3}m$ space group, Fig. 2c). The microspheres thus synthesized exhibited a classical type IV nitrogen adsorption-desorption isotherm with an H2-type hysteresis loop, characteristic of cage-like mesostructure (Supplementary Fig. 4a). The pore size calculated by BJH model showed a spherical cage diameter of approximately 13.9 nm, with the window size between cages measured less than 4 nm (Fig. 2d).

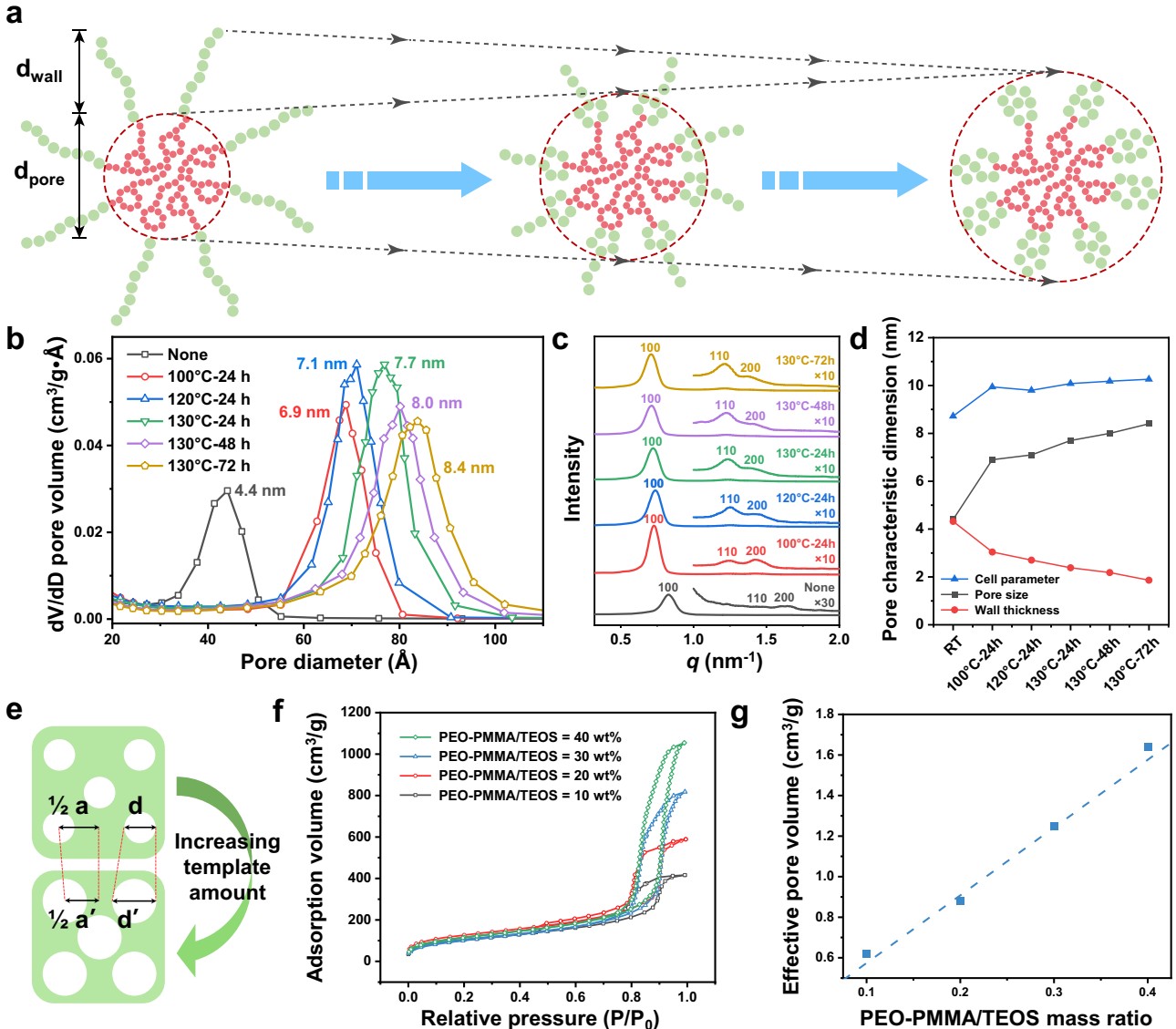

**Fig. 3 | Free tuning of pore structure parameters in monodisperse microspheres with ordered mesopore structure. a** Schematic of pore regulation process by adjusting hydrothermal conditions. **b** Pore size distributions, **c** SAXS patterns, and **d** pore characteristic dimension statistics of ordered mesoporous microspheres under different hydrothermal conditions. **e** Schematic of pore regulation process by increasing the volume of SDAs introduced. **f** Nitrogen adsorption-desorption isotherm and **g** effective pore volume of ordered mesoporous microspheres templated by different PEO-PMMA volumes. The effective pore volume was denoted as the cumulative volume of pores with diameters >2 nm.

Using high-molecular-weight block copolymer, PEO-*b*-PMMA as the SDA, fcc configuration can be synthesized with $F m\bar{3} m$ mesostructure and large unit cell (Fig. 2f), as well as uniform and periodic cage-like pore frameworks (Fig. 2g). The nitrogen adsorption-desorption isotherm demonstrated a steep capillary condensation in the relative pressure from 0.89 to 0.95 and an H1-type hysteresis loop, indicating the generation of uniform large mesopores (Supplementary Fig. 4b). Calculated from adsorption and desorption branches based on BJH model, the pore size reached as high as 20.9 nm, while the window size was approximately 12 nm (Fig. 2h).

One-step microfluidic synthesis of cdg configuration, which is one of the most complex mesostructures, can be challenging. Nevertheless, by first synthesizing silica microspheres with a 2D-hex configuration, followed by a solvothermal post-treatment (80 °C for 72 h in hexane), the mesostructure can be converted into cubic double gyroid ($Ia\bar{3}d$ space group, Fig. 2j). After calcination, the silica microspheres exhibited long-range ordered pore structures (Fig. 2k), distinct from the original 2D-hex mesostructure, with a pore size of 8.4 nm (Fig. 2l and Supplementary Fig. 4c).

Apart from pore configurations, the pore structure parameters of silica microspheres—including pore size, specific surface area, and pore volume—can be finely tuned by adjusting the hydrothermal conditions and the volume of SDA introduced. With silica microspheres templated by P123 as an example (Fig. 3a), the nitrogen adsorption-desorption experiment revealed that the pore size can be tuned from 4.4 to 8.4 nm with the increase of hydrothermal temperature and treatment time, at a pore size adjustment precision of 0.2 nm (Fig. 3b). This angstrom-level pore size was found reproducible across three independent batches, with a CV of 2.1%, demonstrating good reproducibility of pore size tuning via hydrothermal treatment (Supplementary Fig. 8). The full width at half maximum (FWHM) of the mesopores varied from 1.1 to 1.5 nm, showing consistent narrow size distribution across different hydrothermal conditions. The tunability of mesopore size is based on a

general property of non-ionic surfactants: the PEO units undergo partial dehydration at higher temperature, which leads to the volume reduction of the hydrophilic corona and the increase in the micelle size (Fig. 3a). The enlargement of micelle size manifested in two aspects: on one hand, scattering peaks shifted to the larger *d*-spacing, as observed in SAXS spectra (Fig. 3c), accompanied by an increase in the unit cell parameter from 8.72 to 10.26 nm (Fig. 3d); on the other hand, the specific surface area and pore volume were proportionally increased by the micelle size. As presented in Supplementary Table 1, the specific mesopore surface area increased from 410 to 596 m$^2$/g with prolonged hydrothermal treatment time and elevated temperature, meanwhile the pore volume increased significantly from 0.37 to 0.97 cm$^3$/g. The pore structure parameters of large mesoporous cubic $Fm\overline{3}m$ silica microspheres, templated by PEO-*b*-PMMA, exhibited similar hydrothermal regulation effects (Supplementary Table 2).

The effect of SDA volume on pore structure parameters has also been investigated. As shown in Supplementary Fig. 9 and Supplementary Table 3, with the mass ratio of P123/TEOS (marked as P/T) raised from 29.5 to 46.3 wt%, the silica microspheres' isotherms increased in uptake capacity in a stepwise manner. Meanwhile, the pore volume resulted from mesopores showed a linear increase from 0.65 to 1.08 cm$^3$/g, while the pore size expanded from 6.1 to 9.2 nm accordingly. Calculated from *d*-spacing (Supplementary Fig. 10), the unit cell parameter increased from 9.9 to 10.5 nm when P/T increased from 29.5 to 33.7 wt%, and maintained constant between 33.7 to 46.3 wt% P/T. Based on the observed changes in lattice parameters and pore size, it is hypothesized that with the increase of template amount, the micelle size increased first and quickly reached an equilibrium. Further increasing the template amount only changed the hydrophobic portion of micelles, which consequently led to a gradual increase in pore diameter and pore volume, and a reduction in pore wall thickness (from 3.8 to 1.3 nm), as described in Fig. 3e and Supplementary Table 3. The mesoporous silica microspheres templated by PEO-*b*-PMMA also exhibited similar regulation effects (Fig. 3f, g and Supplementary Table 4). However, it is worth noting that, insufficient or excessive amounts of templates can lead to phase transition or disordered micelle assembling, resulting in significant deviations in pore structure parameters from the above-mentioned trends (as evidenced Supplementary Figs. 9, 10 and Supplementary Table 3).

It needs to be highlighted that, during fine-tuning of the internal mesoporous structure, the external morphology of microspheres remained precisely regulated by the droplet templates. As shown in Supplementary Figs.11 and 12, while tuning pore structures, all silica microspheres exhibited both monodisperse size distribution and well-defined spherical morphology. Besides, the effect of SDA volume on the microspheres' sphericity is strongly correlated with the symmetry dimensionality of the ordered mesoporous structure. For instance, in microspheres with 3D symmetric mesoporous configurations—such as fcc templated by PEO-*b*-PMMA—the external morphology remained unchanged while increasing SDA volume, maintaining a good sphericity. In contrast, microspheres with 2D symmetric mesoporous configurations—such as the 2D-hex mesostructure—underwent a significant morphology transition from spherical to polyhedral, and eventually spindle-like, as the introducing amount of P123 increased. On the other hand, the internal mesoporous structure also remained independently regulated by the colloidal templates. As shown in Supplementary Fig. 13, while the silica microsphere size was tuned from 3 to 5 μm using droplets of different sizes as templates, the pore configuration and pore size, pore volume, and specific surface area all remained consistent, which were independent of the outer morphology (Supplementary Fig. 14 and Supplementary Table 5). These results demonstrated that the template-in-template assembly nanostructuring strategy can enable precision and independent regulation of external morphology and internal pore structure, providing a versatile platform for rational design and manufacture of mesoporous microspheres.

In the development of separation methods for multicomponent molecular systems, resolving critical pairs—the two components in a mixture with the lowest resolution[36]—is always a challenging task. The core parameter for evaluating the separation performance of a critical pair is resolution (*Rs*), as defined by Eqs. 1–3[37,38]. Baseline separation is quantitatively defined as Rs ≥ 1.5. While increasing the selectivity factor (α) by stationary phase modification may be an effective way to improve resolution, it involves stationary phase chemistry development, optimization and screening[39,40], and inevitably accompanied by changes in the retention behavior of other components and/or the introduction of unexpected secondary interactions[41]. Therefore, increasing the retention factor (*k*) and separation efficiency (*N*) becomes crucial for pushing limit of the resolution of the critical pairs. According to separation chemistry theory, both *k* and *N* are highly associated with the structure of the packing material, including particle size, size distribution, surface area and pore architecture. As demonstrated above, the TiTAN strategy makes it possible to rationally design and precision manufacture of the nanostructured microsphere materials, which provides an enabling way to simultaneously optimize *k* and *N* to reach an ideal resolution for critical pairs and within a reasonable time frame. For this purpose, we developed monodisperse silica microspheres with 2D-hex mesoporous structure as a separation medium. Such a nanostructured material possesses straight pore channels and uniform pore size due to the 2D-hex configuration, which confers a synergetic advantage: unified diffusion pathways enhancing mass transfer of both ligand molecules during chemical bonding, and analyte molecules during chemical separation. As shown in Supplementary Fig. 15, the performance comparison of different ordered mesoporous structures demonstrated that, as expected, the 2D-hex structure indeed exhibited high retentivity and efficiency.

$$Rs = \frac{\sqrt{N}}{4}(\alpha - 1)\frac{k}{k+1} \qquad (1)$$

$$N = \frac{L}{H} \qquad (2)$$

$$H = A + B/u + Cu \qquad (3)$$

Equations 1–3: where *Rs* is resolution, *N* is separation efficiency, α is selectivity factor, *k* is retention factor; *L* is column length, *H* is theoretical plate height, *u* is linear velocity, *A*, *B*, and *C* are numerical coefficients related to the parameters of the column.

The TiTAN synthesized silica microspheres exhibited a monodisperse size distribution with a mean diameter of 5.0 μm (CV = 4.7%, Supplementary Fig. 16). After functionalization with monomeric C18 ligands, the microspheres were packed into a 100 μm i.d. column (denoted as OMP, for ordered mesoporous particle). The OMP column demonstrated high mechanical stability under prolonged high-pressure conditions (65 MPa), maintaining structural integrity without particle fragmentation or bed deformation (Supplementary Figs. 17 and 18). To evaluate the chromatographic performance of the OMP column, a reference column packed with conventional porous particles (denoted as TPP, for totally porous particle) of identical size (5.0 μm mean diameter, CV = 11.8%), column dimensions and bonding stationary phase was used as a benchmark for comparison.

Separation performance was evaluated under isocratic conditions (60:40 v/v ACN/H$_2$O) using a standard mixture of thiourea (dead-time marker) and alkylbenzenes (methyl-, ethyl-, propyl-, and butylbenzene). As shown in Fig. 4a, the OMP column exhibited significantly stronger retention than the TPP column: butylbenzene eluted with ~40

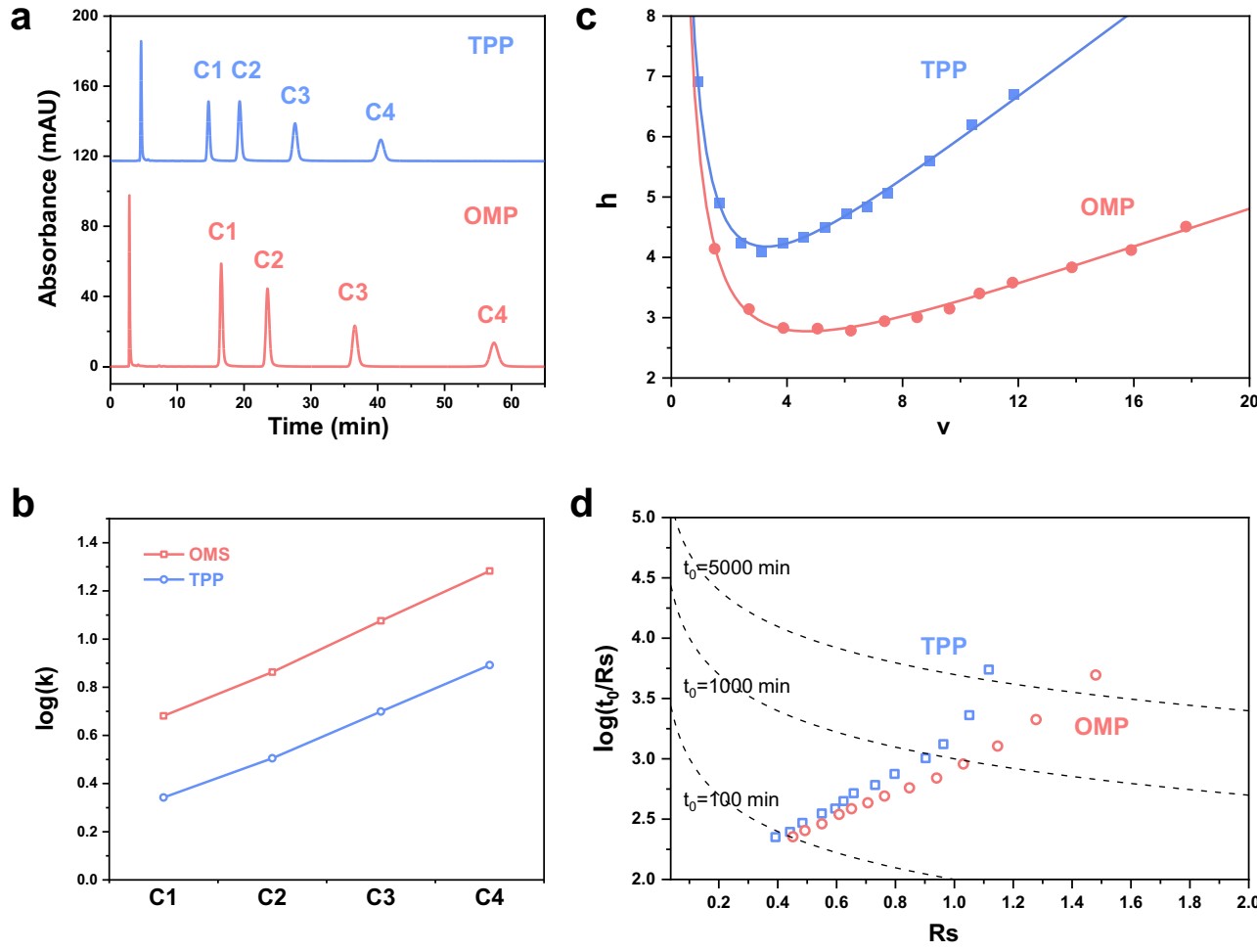

**Fig. 4 | Performance comparison between ordered mesoporous particle and totally porous particle. a** Chromatograms of an alkylbenzene mixture on 15 cm long OMP and TPP columns. Analytes: thiourea, methyl-, ethyl-, propyl-, and butylbenzenes (in order of elution); mobile phase: 60:40 v/v ACN/H₂O; UV detection: 214 nm. **b** Comparison of retention factor, **c** Knox curves, and **d** modified kinetic plots of OMP and TPP.

% longer retention time on OMP. The retention factor on OMP increased more steeply with analytes' hydrophobicity, reaching $k = 19.2$ (butylbenzene) on the OMP column versus $k = 7.8$ on the TPP column (Fig. 4b). As shown in Supplementary Table 6, elemental analysis and nitrogen adsorption-desorption experiment revealed that OMP supports a higher C18 ligand density (2.65 μmol/m²) compared to TPP (<2.30 μmol/m²), and offers a larger specific surface area (500 m²/g versus 350 m²/g), respectively. The combination of increased ligand coverage and accessible surface area concordantly raised the phase ratio, enhancing the retentivity of the ordered mesoporous separation material. The raised phase ratio also resulted in a high loading capacity of the OMP material, as demonstrated in the loading mass test (Supplementary Fig. 19).

Figure 4c presented the reduced plate height ($h$) versus reduced linear velocity ($v$) for butylbenzene on the OMP and TPP columns. Without correction for extra-column band broadening, the minimum reduced plate height ($h_{min}$) is 2.78 for the OMP column—only 68 % of that of the TPP column ($h_{min} = 4.08$), representing almost 50% enhancement of efficiency. The coefficients a, b, and c (reduced form of A, B, and C, respectively) are listed in Supplementary Table 7, by fitting the data to Knox equation (Eqs. 4–6 as detailed in "Methods"). The a-term for the OMP column is merely 62% of that for TPP column, indicating that the narrowed particle size distribution (due to the monodisperse spherical morphology) significantly reduces eddy-

diffusion, the major contribution of band broadening. Despite a smaller average pore diameter (Supplementary Table 6), the OMP column still exhibited a reduced c-term (by 56%) than the TPP column. This demonstrated that the ordered 2D-hex pore configuration effectively reduced intraparticle mass-transfer resistance and accelerated partition equilibrium of the analytes between the mobile and stationary phases, leading to the pronounced enhancement of separation efficiencies.

An additional set of totally porous particles (denoted as hTPP, for homemade totally porous particle) was intentionally synthesized to closely matched to OMP in terms of particle size, monodispersity, and pore size (Supplementary Fig. 20 and Supplementary Table 8). As shown in Supplementary Table 7 and Supplementary Fig. 21, OMP exhibited pronouncedly higher efficiency (2.78 vs. 3.82 for $h_{min}$) and higher retentivity (19.2 vs. 4.0 for $k$) than that of hTPP, both outperforming the conventional totally porous counterparts.

Benefiting from monodisperse particle size distribution, higher accessible surface area, and straight and uniform pore configuration, both $k$ and $N$ of the OMP column increased, and the $Rs$ for critical pairs can be improved as a natural result. To quantitatively evaluate the resolution enhancement, a modified kinetic plot was constructed (as detailed in "Methods"). A hypothetical critical pair whose selectivity factor set as low as 1.01 was chosen as probing compounds, while the column efficiency was kept same. As shown in Fig. 4d, calculations

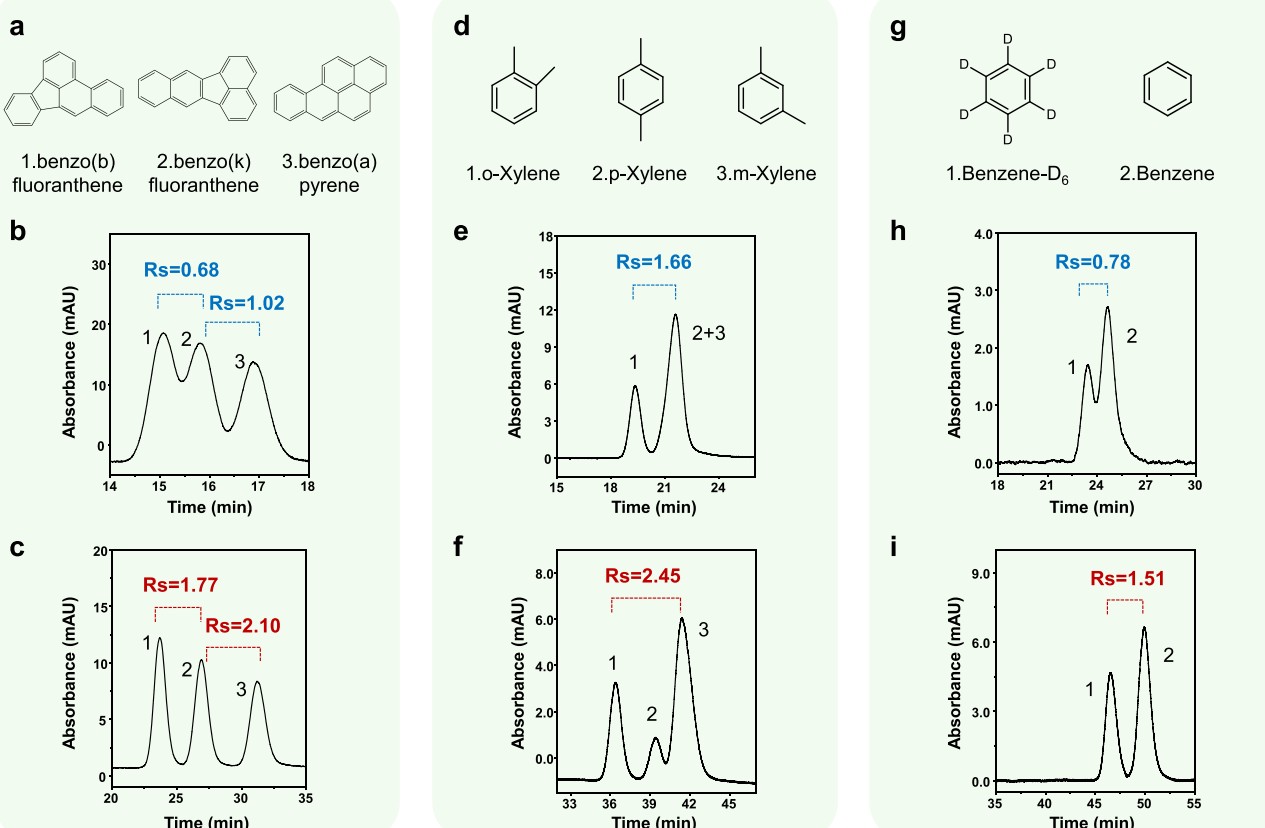

**Fig. 5 | Separations of critical pairs on monodisperse microspheres with ordered mesoporous structure.** Chromatograms of **a** a polycyclic aromatic hydrocarbon mixture on **b** TPP and **c** OMP columns. Mobile phase: 70:30 v/v ACN/H$_2$O. Chromatograms of **d** a xylene isomer mixture on **e** TPP and **f** OMP columns. Mobile phase: 50:50 v/v ACN/H$_2$O. Chromatograms of **g** benzene/benzene-d$_6$ on **h** TPP and **i** OMP columns. Mobile phase: 20:80 v/v ACN/H$_2$O. All separations were performed on 15 cm columns under isocratic elution conditions at a flow rate of 800 nL/min with UV detection at 210 nm.

show that the resolution achievable on the OMP column is higher than that on the TPP column over the entire time range. For short time runs (<100 min), the OMP column's resolution is slightly higher; however, as the analysis time increases and more theoretical plates are generated, the advantage of the OMP column becomes increasingly significant. Specifically, the OMP column achieves a 14% higher $Rs$ than TPP column at 1000 min, and the $Rs$ enhancement becomes 33% at 10,000 min. From a separation speed point of view, in order to reach the same $Rs$ (e.g., $Rs = 1.1$), the conventional TPP column needs 4380 min, whilst the OMP column needs only 1178 min, ~3/4 shortened run time, indicating a significantly faster separation speed while realizing the same level of resolution, by using ordered mesoporous separation materials.

To demonstrate the potential of the precisely constructed ordered mesoporous separation media for high performance chromatography, a series of challenging critical pairs was chosen for investigations. Polycyclic aromatic hydrocarbons (PAHs) are a class of homologous compounds composed of multiple benzene rings with similar molecular weights. Due to the absence of π–π interaction or spatial stereoselectivity of the classical monomeric C18 phase, achieving effective resolution of PAHs solely based on the hydrophobicity of C18 chemistry remains a significant challenge[42,43]. On the OMP column, as shown in Fig. 5a–c and Supplementary Fig. 22, sixteen standard PAHs were effectively separated under isocratic conditions. The critical pair, benzo(b)fluoranthene and benzo(k) fluoranthene, realized ideal separation with $Rs = 1.77$, while the TPP column only reached a low resolution of $Rs = 0.68$. Impressively, indeno(1,2,3-cd)pyrene and benzo(g,h,i)perylene have been resolved

on the OMP column with $Rs = 0.92$, while on the TPP column the critical pair cannot be distinguished at all.

Isomers, such as xylene, possess identical molecular weights and almost identical hydrophobic characteristics (Δlog P ≈ 0.08 for xylene)[44], with merely slight difference in dipole moment and spatial configuration[45]. In previous studies, efficient separation of xylene isomers has required the incorporation of additional interactions to improve selectivity, or the use of materials with pore sizes matching isomers dimensions, such as molecular sieves or MOFs, for spatial resolution[46–49]. In this study (Fig. 5d–f), under isocratic elution (50:50 v/v ACN/H$_2$O) on the OMP column, o-xylene was effectively baseline-separated from p- and m-xylene with $Rs = 1.78$ and 2.45, respectively, while p- and m-xylene also realized resolution ($Rs = 0.96$). In contrast, running the mixture on the TPP column, only o-xylene can be resolved, while no resolution was identified for the p- and m-xylene.

Isotopologue separations—where analytes differ only in constituent element weight, and their dipole moment and hydrophobicity are virtually indistinguishable—represent one of the most stringent tests of chromatographic resolution[50,51]. Typically, efforts on separating such pairs have required either extraordinarily long columns or recycling chromatography[52,53] in order to generate sufficiently high numbers of theoretical plates, which is a time-consuming and labor-intensive process. As shown in Fig. 5g–i, with OMP medium, however, under isocratic conditions (20:80 v/v ACN/H$_2$O), the 15 cm short OMP column achieved a satisfactory $Rs = 1.51$ for benzene/benzene-d$_6$ within 55 min, whereas the TPP counterpart delivered only partial separation ($Rs = 0.78$). For the even more challenging dibromobenzene/dibromobenzene-d$_4$ pair (Supplementary Fig. 23), where the TPP column

failed to recognize the isotopologues at all, the OMP column, within 250 min, successfully split the critical pair ($Rs = 0.67$).

## Discussion

Generally, almost all of the well-developed synthetic strategies for separation media can only achieve controllability of either material morphology or pore structure, but not the both. This limitation arises from synthesis methodologies constraints. On one hand, the orderliness of pore structure is formed through periodic arrangement of porogenic units/templates, which limits radial symmetric growth required for uniform spherical morphology. This assembly mechanism predominantly results in either randomly aggregated bulk materials or crystalline particulates. On the other hand, materials with uniform morphology are primarily assembled by layer-by-layer deposition/growth/swelling processes, or directly manufactured by ultrahigh-precision microfabrication techniques (e.g., silicon micromachining[54] or 3D printing[55,56]). However, these approaches are generally incompatible with the synthetic systems required for forming ordered nanostructures, typically yielding materials with disordered porous networks. Although a few strategies can obtain microspheres with uniform pore structure[57,58], the pore configurations realizable were typically limited, without the flexibility to modulate pore size and configuration. In this work, we present a microfluidic-based TiTAN methodology that effectively addresses this long-standing synthetic dilemma. The morphological characteristics are precisely regulated using microfluidic droplets as shape-directing templates, while the pore structure is configured through the self-assembly of porogenic templates within the droplet templates. By synchronizing the sol-gel reaction kinetics with the template assembly process, we achieve independent and flexible control over these two critical structural chemistry features, enabling precise architecture of both pore structure and sphere morphology.

Some works have attempted to synthesize porous microspheres via droplet microfluidics[59,60], nevertheless, these strategies suffered from poor versatility of the synthesis systems, limited controllability of particle morphology, and low production throughput, which constrained the application in large-scale manufacturing of separation media. The current strategy addresses these limitations through a fluorinated oil-based droplet generation system. First, the hydrophobic and lipophobic nature of fluorinated oil allows stable encapsulation of various organic volatile solvents, precursors and hydrophobic porogenic templates within the dispersed phase, while maintaining high generation throughput of monodisperse droplets (Supplementary Fig. 24 and Supplementary Movie 2). This capability not only facilitates the synthesis of silica microspheres with diverse ordered mesoporous structures, but also extends the types of material chemistry from silica to organosilica and transition metal oxides, such as ethane-bridged silica (Supplementary Figs. 25–27), phenyl-bridged silica (Supplementary Figs. 27–29), titanium dioxide (Supplementary Figs. 30 and 31) and zirconium dioxide (Supplementary Figs. 32 and 33). Second, the exceptionally high density of fluorinated oils promotes spontaneous droplet flotation at the solution surface, which enables controlled solvent evaporation under mild conditions. Consequently, conditions for ordered mesopore formation can be precisely adjusted and optimized.

The TiTAN synthesized microspheres presented high performance in both kinetic properties and practical separations. This high performance can be attributed to three key structural advantages: First, the well-defined spherical morphology and uniform size distribution effectively reduce eddy diffusion, the major contribution to band broadening, thereby enhancing separation efficiency. Second, despite having smaller pore size than commercial TPP separation media, OMP exhibited reduced mass-transfer resistance under identical conditions, which demonstrates that the ordered pore architecture can streamline molecular diffusion within the pores,

contributing to improved separation efficiency. Third, the ordered pore structure substantially increases the accessible surface area, enabling a high bonding density and consequently greater retentivity. Consequently, the precisely controlled morphology and pore architecture endow OMP media with enhanced resolving power, achieving baseline separations of analytes that were either unresolvable or barely resolving using conventional separation media, and within a reasonable time frame. The chromatographic performance of the orderly nanostructured microspheres reported in this work demonstrated the importance of precise morphological and structural control in separation media design, and also highlight the significant potential of the TiTAN strategy in supporting advanced separation applications.

Taking the long-standing challenge of simultaneous controllability over both spherical morphology and pore architecture in separation media, we developed a TiTAN strategy that enables precision synthesis of chromatographic microspheres with uniform morphology and ordered mesoporous structures at high flexibility and versatility. By utilization of fluorinated oil-based droplet microfluidic, the TiTAN synthesis strategy enables independent regulations of morphological characteristics and pore structure. Through precise design of nanostructures, separation media with various ordered mesoporous architectures can be configured, including 2D hexagonal, centered cubic, face centered cubic and cubic double gyroidal structures, while maintaining excellent monodispersity and sphericity with tunable sphere size and material chemistry. Furthermore, by adjusting the volume of porogenic templates and hydrothermal treatment conditions, we achieved precise control over pore structure parameters (surface area, pore size, and pore volume) with a spatial resolution as fine as 2 Å. The combination of monodisperse spherical morphology and ordered pore architecture endows the separation media with significantly enhanced efficiency and retention, leading to high chromatographic resolution. This advancement enables effective separation of particularly challenging critical pairs with low selectivity factors, including polycyclic aromatic hydrocarbons, xylene isomers, and benzene-isotopologue molecules. Moreover, the flexible and broad tunability of pore structures in these materials provides a powerful platform for investigating the influence of pore architecture on chromatographic kinetics, as well as for rational design and precision manufacture of separation media for diverse applications, ranging from small-molecule, pharmaceutical to biomacromolecule separations.

## Methods

### Synthesis of ordered mesoporous silica microspheres

The ordered mesoporous silica microspheres were synthesized based on the TiTAN strategy. Typically, the continuous phase was fluorinated oil-HFE-7500 with 1 wt% PFPE-*b*-PEG-*b*-PFPE as surfactant, and a sol-gel reaction precursor containing SDAs was applied as the dispersed phase. The microfluidic platform enabled droplet generation at a super throughput of 240,000 Hz per chip, resulting in a production yield of approximately $10^9$ microspheres per hour.

To prepare the sol-gel reaction precursor, 400 μL TEOS was hydrolyzed in 268 μL HCl (0.1 M) under vigorous stirring at 1000 rpm for 60 min. Then, the hydrolyzed TEOS solution was mixed with a porogen solution tailored to the target pore configuration. For 2D-hexagonal configuration, the porogen solution was 1890 μL acetonitrile containing 145 mg P123. For cubic double gyroidal configuration, the porogen solution was 1822 μL acetonitrile containing 190 mg P123. For body centered cubic configuration, the porogen solution was adjusted to 1821 μL acetonitrile containing 161 mg F127. For face centered cubic configuration, the process required initial hydrolysis of 400 μL TEOS in 750 μL HCl (0.3 M) for 30 min before mixing with a porogen solution consisting of 74.4 mg PEO-PMMA and 6215 μL acetonitrile.

For droplet generation, the dispersed phase (36 μL/min) and continuous phase (120 μL/min) were injected into the microfluidic chip by syringe pumps. The generated droplets were collected in a petri dish containing FC-40 fluorinated oil, and then subjected to solvent evaporation and solidification for 12 h at 25 °C. The solidified microspheres were transferred to an autoclave containing 1 M HCl solution and placed in an oven for 24 h at 130 °C. After that, the microspheres were calcined in air at a rate of 1 °C/min to 550 °C, and kept at 550 °C for 6 h to completely remove SDAs. For certain configurations, the hydrothermal treatment varied slightly –80 °C treatment for 72 h in hexane for cubic double gyroidal configuration, and 80 °C treatment for 24 h in hexane for body centered cubic configuration.

### Synthesis of ordered mesoporous microspheres with other substances

For the synthesis of ethane-bridged silica microspheres, the dispersed phase was prepared by first mixing 330 μL 1,2-bis(triethoxysilyl)ethane with 268 μL HCl (0.1 M) under stirring for 1 h, followed by addition of 1820 μL acetonitrile containing 144 mg P123. After hydrothermal treatment, the SDAs were removed by Soxhlet extraction with ethanol at 80 °C for 24 h. Phenyl-bridged silica microspheres were synthesized by mixing 153 mg of 1,4-bis(triethoxysilyl)benzene with 150 mg P123, 300 μL HCl (0.1 M), and 1350 mg methanol under stirring for 5 min, the SDAs were also removed by Soxhlet extraction with ethanol at 80 °C for 24 h. Titania microspheres were prepared by mixing 400 μL titanium (IV) butoxide, 120 μL acetylacetone, and 200 μL HCl (2 M) under stirring for 5 min, followed by addition of 2180 μL acetonitrile containing 122 mg P123. After droplet solidification, titania microspheres were calcined in air at a rate of 1 °C/min to 350 °C, and kept at 350 °C for 6 h to totally remove SDAs. Zirconia microspheres were obtained by mixing 564 mg zirconium (IV) butoxide, 120 μL acetylacetone, 800 μL ethanol, and 400 μL HCl (2 M) under stirring for 30 min, followed by addition of 2180 μL acetonitrile containing 122 mg P123, the calcination conditions were identical to that of titania microspheres.

### Stationary phase bonding

The TiTAN-synthesized silica microspheres were bonded with octadecylsilyl stationary phase. Specifically, 50 mg microspheres were dried at 120 °C for 12 h. Then, 0.12 mmol dimethyloctadecyl-chlorosilane and 0.24 mmol 2,6-lutidine were dissolved in 5 mL dry toluene. Dry microspheres were dispersed in the mixed solution and reacted under $N_2$ at 80 °C for 40 h. Finally, the microspheres were thoroughly washed with dichloromethane, methanol, and water.

### Chromatographic column preparation

TiTAN-synthesized microspheres and commercial packing materials were chosen to pack capillary liquid chromatography columns. The capillary columns were prepared by high-pressure slurry packing method[61,62]. Briefly, a single perfusive particle with ~100 μm in diameter was tapped into one end of the capillary (100 μm i.d. and 365 μm o.d.) to be packed, which was used as the outlet frit of the column. Then, from the other end of the capillary, a slurry of the packing material was introduced and packed under a high pressure up to 6000 psi. When the packing was finished, another single particle was tapped into the inlet of the capillary.

### Evaluation of chromatographic performance

Chromatographic columns were evaluated on an Ultimate 3000 RSLCnano liquid chromatography system (Thermo Scientific). Alkylbenzenes were used as samples to evaluate separation performance (toluene, ethylbenzene, propylbenzene and butylbenzene), and thiourea was added to determine the column dead-time. The samples were injected by a 4 nL Valco nanovolume injector, eluted under the isocratic condition with a mobile phase of 60/40 v/v ACN/H2O, and detected by a UV detector at the wavelength of 214 nm. The Knox

curve was constructed to eliminate the effect of particle size[63,64], which can be described as:

$$h = av^{\frac{1}{3}} + \frac{b}{v} + cv \qquad (4)$$

The reduced plate height ($h$), and reduced linear velocity ($v$), were calculated by:

$$h = \frac{H}{d_p} \qquad (5)$$

$$v = \frac{ud_p}{D_m} \qquad (6)$$

Where $d_p$ is the particle size of the packing material, and $D_m$ is the diffusion coefficient of the analyte. The diffusion coefficient of butyl-benzene referred to the database established by the previous study[65].

Additionally, a modified version of the kinetic plot was constructed to quantitatively evaluate the separation resolution potential. Typically, the standard kinetic plot was obtained by transforming a series of experimentally measured ($u_0$, $H$)-data into a set of ($t_0$, $N$)-data[66–68], using Eqs. 7–8:

$$N = \frac{\Delta P_{max} K_{v0}}{\eta u_0 H} \qquad (7)$$

$$t_0 = \frac{\Delta P_{max} K_{v0}}{\eta u_0^2} \qquad (8)$$

Where $\Delta P_{max}$ is maximal allowable instrument pressure. $K_{v0}$ is the permeability, $\eta$ is mobile phase viscosity, $u_0$ is linear velocity and $H$ is theoretical plate height (calculated based on butylbenzene). In this work, the kinetic plot was constructed for a pressure limit of 60 Mpa.

Further, the modified kinetic plot can be obtained by transforming ($t_0$, $N$)-data into a set of ($\log(t_0/Rs)$, $Rs$)-data, replacing $N$ with $Rs$ using Eq. 1 (as already introduced in the main text). Wherein, a hypothetical critical pair whose selectivity factor ($\alpha$) set as low as 1.01 was chosen as probing compounds, while the column efficiency was kept same. The retention factor ($k$) of the hypothetical critical pair was equal to that of butylbenzene (19.2 for OMP and 7.8 for TPP).

### Separation of critical pairs

All chromatographic separations were performed under isocratic elution conditions at a flow rate of 800 nL/min with UV detection at 210 nm. For polycyclic aromatic hydrocarbons separation, 70/30 v/v ACN/H2O were applied as the mobile phase. For xylene isomers separation, 50/50 v/v ACN/H2O were applied as the mobile phase. For isotopologues separation, 20/80 v/v ACN/H2O were applied as the mobile phase.

## Data availability

The data supporting the findings of this study are available within the paper. Data is available from the corresponding author on request. Source data are provided with this paper.

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

## Acknowledgements

We thank Prof. Yukui Zhang of Dalian Institute of Chemical Physics, Chinese Academy of Science, for valuable discussion. B.Z. acknowledges financial support from the National Key Research and Development Program of China (2023YFF0713900), National Natural Science Foundation of China (22574134, 21475110), Scientific Research Foundation of State Key Laboratory of Vaccines for Infectious Diseases, Xiang An Biomedicine Laboratory (2025XAKJ0203002), NFFTBS (J1310024) and PCSIRT (IRT_17R66).

## Author contributions

J.Z., B.Z., and Z.Z. conceived the project and designed the experiments. J.Z., B.Z., H.C., and K.S. co-wrote the manuscript. J.Z., J.C., and X.H. carried out the synthesis and characterization of the materials. H.C., L.L., and X.W. were involved in the application data collection. Z.Z., H.C., and K.S. assisted J.Z. with the data collection and analysis. All authors contributed to the discussion and manuscript preparation.

## Competing interests

The authors declare no competing interests.
