## [Transparent Peer Review file · Nature Communications]

Template-in-Template Assembly Nanostructured Microspheres for High Performance Chromatography

Corresponding Author: Professor Bo Zhang

Version 0:

Reviewer comments:

Reviewer #1

(Remarks to the Author)

The authors present a novel strategy termed template-in-template assembly nanostructuring (TiTAN) for the precise synthesis of monodisperse microspheres with ordered mesoporous nanostructures. They further demonstrate the potential application of these materials in high-performance chromatography. While the work builds upon the authors' prior microfluidic precision manufacturing approach (Angew. Chem. 2024), its key innovation lies in the simultaneous architectural control over both morphology and nanostructure in mesoporous microspheres. From a fundamental nanoscience perspective, this represents an elegant and technically sound advancement.

However, the claimed benefits of such mesoporous nanostructures for chromatographic separation remain unconvincing for several reasons:

Limited Separation Efficiency: The plate height values (Figure 4C) indicate suboptimal separation efficiency, suggesting that the mesoporous nanostructure does not confer meaningful improvements in chromatographic kinetics.

Lack of Demonstrated Selectivity: The separations shown in Figure 5 fail to achieve baseline resolution for the tested analytes, raising doubts about any substantive enhancement in selectivity.

Durability Concerns: The mechanically fragile nature of the surface mesopores casts uncertainty on the long-term stability and applicability of these materials for complex or high-throughput separations.

Given these limitations, the work, while methodologically interesting, does not presently meet the threshold for transformative impact expected by high-profile interdisciplinary journals such as Nature Communications.

Minor Comments:

Resolution Calculation (Figure 5b2): The reported resolution value of 1.66 appears inconsistent with the lack of baseline separation in the chromatogram. The authors should verify their calculations.

Improved Data Representation: A plot of $\log k$ versus carbon number would better assess whether the separation follows a reversed-phase mechanism, as such a relationship should yield a linear trend.

Reviewer #2

(Remarks to the Author)

The manuscript by Prof. Zhang and coworkers reports the synthesis of monodisperse microparticles with a mesoporous nanostructure for chromatographic applications. The preparation of silica microparticles begins with the use of microfluidics to generate monodisperse droplets containing TEOS, hydrochloric acid, and an amphiphilic polymer surfactant. Subsequently, rapid solvent evaporation from the droplets induces self-assembly of the surfactant into a liquid crystalline phase, while simultaneously promoting the condensation of silica in the hydrophilic domains. Finally, hydrothermal

treatment and calcination yield monodisperse silica particles with an ordered porous structure. By adjusting the SDAs and post-treatment conditions, the authors demonstrate that the nanostructures of the microparticles can be finely tuned. The authors further showcase the application of these particles in chromatographic separation, where they exhibit high performance compared to conventional silica particles.

In my opinion, this work is very interesting; however, several issues still need to be addressed prior to publication:

1. Almost all of the SEM images lack surface details, making it difficult to extract useful information beyond particle size from the presented images.
2. Why did the authors choose to use the particles mentioned in the main text (CV = 4.7%, Figure S14) for the separation performance comparison, rather than the 5 μm particles shown in Figure S12, which clearly have a more spherical morphology and more uniform size?
3. In the comparison of separation performance, there are too many variables between the two types of particles, such as uniformity, pore structure, and morphology, making it difficult to draw accurate conclusions.
4. The authors are encouraged to compare the influence of different nanostructures on separation performance to provide more comprehensive conclusions.
5. It would be helpful for the authors to clarify the production yield of such porous particles prepared by microfluidics, as well as the sample loading capacity that can currently be separated using their chromatographic system.

Reviewer #3

(Remarks to the Author)

The authors have presented a method for synthesizing mesoporous particles with a well-defined internal structure using a microfluidic method they refer to as TiTAN. This method is especially exciting for uses in chromatography, as the authors have demonstrated in their measurements showing superior separations of small molecules using particles with different internal structures and similar macroscopic ($\sim 4 \mu\text{m}$) sizes. In my opinion, these are significant results in terms of the synthesis and application of the particles in an incredibly useful way -- and likely to be highly impactful. In fact, I can think of several more applications of these particles to fundamental scientific studies that extend beyond chromatography.

Nevertheless, as I am not an expert in chromatography, my review of this manuscript focuses exclusively on the authors' claims regarding the well-defined internal structure of the particles as probed using small-angle X-ray scattering (SAXS) and gas adsorption measurements. SAXS is one of the most appropriate techniques for measuring structure in systems such as this. Gas adsorption measurements similarly provide a good estimate of the pore size, and demonstrate the authors can tune this parameter with great precision. I have a few questions and concerns regarding these results, although overall I believe they strongly support the authors' conclusions.

(1) A comment: I would appreciate if the authors would label the relative peak positions (in terms of scattering vector Q) relative to the fundamental peak (Q^*) in their figures, rather than indexing the reflections. While this information is redundant, it would help the reader interpret these results. The authors should indicate each peak perhaps with an arrow of some sort. I realize this request may seem odd, but indexing the peaks this way makes it much easier for the reader to assess the quality of the mesopore structure. This is commonly done in the block copolymer community (see for example Chapter 5 in Basic X-ray Scattering for Soft Matter by de Jeu).

(2) In some samples, for example the Ia-3d/gyroid structures, the higher order reflections are quite weak (see Fig. 2g). How are the authors identifying peaks, and can they offer some insight into the relatively weak signals for higher values of Q ?

(3) The authors show extraordinary control over the pore size (Fig. 3b). I assume these results are reproducible, but I wonder if the authors can comment on the variability between pore sizes between independent batches of particles? I may have missed this, but I cannot find in the manuscript or SI how many independent trials were performed. Were the nitrogen adsorption-desorption measurements performed on independent batches? Is the high tunability of pore size that the authors demonstrated within a single batch of the particles? Etc. Some clarity regarding these points is necessary.

Version 1:

Reviewer comments:

Reviewer #2

(Remarks to the Author)

[Note from the Editor: this reviewer looked over responses to their own comments, and the comments from reviewer 1.]

In my opinion, the authors have addressed the concerns from Reviewer #1.

Reviewer #3

(Remarks to the Author)

The authors have fully addressed my comments regarding the original manuscript, and I am fully satisfied with their responses. I am pleased to recommend publication of this work to the Editor.

Response to Reviewer Comments

Reviewer #1:

The authors present a novel strategy termed template-in-template assembly nanostructuring (TiTAN) for the precise synthesis of monodisperse microspheres with ordered mesoporous nanostructures. They further demonstrate the potential application of these materials in high-performance chromatography. While the work builds upon the authors' prior microfluidic precision manufacturing approach (Angew. Chem. 2024), its key innovation lies in the simultaneous architectural control over both morphology and nanostructure in mesoporous microspheres. From a fundamental nanoscience perspective, this represents an elegant and technically sound advancement.

We sincerely appreciate your positive and insightful comments regarding our work. Your encouraging remarks are a strong motivation for our research. In response to your comments, we have made the following revisions:

- (i) We have re-optimized the methods and re-performed the separations to achieve baseline resolution of the critical pairs, and updated the chromatography results accordingly in Figure 5.
- (ii) We have performed ultrahigh pressure experiments to evaluate the mechanical stability of ordered mesoporous particles.
- (iii) We have verified the resolution calculation and clarified the definition of baseline separation in the manuscript.
- (iv) We have also replaced Figure 4b with a plot of $\log(k)$ versus carbon number.

However, the claimed benefits of such mesoporous nanostructures for chromatographic separation remain unconvincing for several reasons:

1. Limited Separation Efficiency: The plate height values (Figure 4C) indicate suboptimal separation efficiency, suggesting that the mesoporous nanostructure does not confer meaningful improvements in chromatographic kinetics.

Response: Thank you for raising this point. We would like to clarify that, among the four ordered mesoporous nanostructures reported in our study, the 2D hexagonal configuration is indeed not the one with the best chromatographic kinetics. Nevertheless, we consider it an ideal packing material for chromatographic investigations. This is because the objective of this work was to design and manufacture well-structured materials towards improved chromatographic resolution; while chromatographic resolution, by definition, is not entirely built upon column efficiency but also retentivity (as shown in Equation 1). Therefore, in selecting the mesoporous configuration for chromatographic evaluations, both efficiency and retentivity were taken into account.

Specifically, as shown in Fig. A1(a), face-centered cubic mesoporous microspheres can achieve an excellent reduced plate height ($h_{\min} = 1.9$), outperforming the 2D hexagonal mesoporous microspheres

($h_{\min} = 2.7$). However, the retention factor of the face-centered cubic mesoporous microspheres ($k = 1.34$) is considerably lower than that of the 2D hexagonal mesoporous microspheres ($k = 19.2$). Taking both into account, the 2D hexagonal nanostructure offers a favorable balance between efficiency and retentivity.

To further justify the selection of the 2D hexagonal mesoporous microspheres for application, we compared the modified kinetic plots of face-centered cubic mesoporous microspheres (high efficiency but low retentivity) and 2D hexagonal mesoporous microspheres (high retentivity with decent efficiency comparable to most commercial columns). As shown in Fig. A1(b), the 2D hexagonal mesoporous microspheres realized higher resolutions in shorter time, making them ideal for separations of challenging critical pairs. At the end of the day, chromatography is about resolution, while efficiency, certainly a key factor, is used to support the total resolution as a collective result.

Figure A1. Comparison of Knox curves (a) and modified kinetic plots (b) of 2D hexagonal mesoporous microspheres (2D-hex) and face-centered cubic mesoporous microspheres (fcc).

2. Lack of Demonstrated Selectivity: The separations shown in Figure 5 fail to achieve baseline resolution for the tested analytes, raising doubts about any substantive enhancement in selectivity.

Response: We appreciate your insightful view. In the application, we intentionally selected the critical pairs with challenging selectivity factors, in order to demonstrate the significant resolution enhancement provided by the ordered mesoporous nanostructures' enhanced retentivity and efficiency. As evidenced in Figure 5d-f, even though the critical pairs are esteemed challenging to resolve chromatographically, the ordered mesoporous materials are still able to provide good selectivity allowing for distinctive recognition of the peaks.

However, we agree with you that achieving baseline resolution for the tested analytes should be substantively meaningful. By definition, when resolution (R_s) is or above 1.5, the peaks are considered baseline separated, allowing for clear identification and individual integration (for quantification), which is of paramount importance to analytical chemists. In this regard, we have re-optimized the method and re-performed the separation. As shown in the revised Figure 5g-i, after re-optimized the mobile phase from 30:70 v/v ACN/H₂O to 20:80 v/v ACN/H₂O, the R_s of benzene and benzene-d₆ on ordered mesoporous particles increased from 1.21 to 1.51, achieving the baseline separation. While under the same conditions, the R_s on totally porous particles increased from 0.65 to 0.78, remained significantly lower than that achievable with the ordered mesoporous particles, let alone baseline separation. We have also updated Figure 5a-c with the baseline-separated chromatogram of benzo(b)fluoranthene, benzo(k)fluoranthene, and benzo(a)pyrene, which was originally put in the Supporting Information. In

total, compared to the totally porous particles, the ordered mesoporous material demonstrated a pronounced improvement for critical pairs from non-baseline to baseline separations.

Changes to the Manuscript, Figure 5:

Figure 5. Separations of critical pairs on monodisperse microspheres with ordered mesoporous structure. (a-c) Chromatograms of a polycyclic aromatic hydrocarbon mixture (a) on TPP (b) and OMP (c) columns. Mobile phase: 70:30 v/v ACN/H₂O. (d-f) Chromatograms of a xylene isomer mixture (d) on TPP (e) and OMP (f) columns. Mobile phase: 50:50 v/v ACN/H₂O. (g-i) Chromatograms of benzene/benzene-d₆ (g) on TPP (h) and OMP (i) columns. Mobile phase: 20:80 v/v ACN/H₂O. All separations were performed on 15 cm columns under isocratic elution conditions at a flow rate of 800 nL/min with UV detection at 210 nm.

Changes to the Manuscript, Line 326: As shown in Figure 5g-i, with OMP medium, however, under isocratic conditions (20:80 v/v ACN/H₂O), the 15 cm short OMP column achieved a satisfactory $R_s = 1.51$ for benzene/benzene-d₆ within 55 min, whereas the TPP counterpart delivered only partial separation ($R_s = 0.78$).

3. Durability Concerns: The mechanically fragile nature of the surface mesopores casts uncertainty on the long-term stability and applicability of these materials for complex or high-throughput separations.

Response: Thank you for emphasizing the importance of mechanical stability. Figure S17 shows SEM images of the ordered mesoporous particles (OMP) before and after pressure testing, demonstrating that the OMP material maintains structural integrity without fragmentation up to at least 50 MPa. However, more quantitative data are needed for this concern. Therefore, we have supplemented the study with two additional experiments: (1) we tested the back pressure as a function of flow rate and extended the maximum pressure test to 65 MPa; and (2) we also evaluated the stability of OMP during

long-term operations under ultra-high pressure. The results from these experiments provide strong evidence for the long-term stability and applicability of OMP materials for complex or high-throughput separations.

Changes to the Supporting Information:

Figure S18. Mechanical stability test of ordered mesoporous particles. (a) Back pressure as a function of flow rate; (b) Back pressure fluctuation during 24 hr running at the flow rate of 1.0 µL/min. Column: 27 cm × 100 µm i.d.; mobile phase: 50:50v/v MeOH/H₂O.

Changes to the Manuscript, Line 241: The OMP column demonstrated high mechanical stability under prolonged high-pressure conditions (65 MPa), maintaining structural integrity without particle fragmentation or bed deformation (Figures S17 and S18).

Given these limitations, the work, while methodologically interesting, does not presently meet the threshold for transformative impact expected by high-profile interdisciplinary journals such as Nature Communications.

Response: Based on your insightful and constructive comments, we have performed further experiments and extended the discussion in order to solve all the concerns you raised. We believe our detailed responses and the revised Manuscript and Supporting Information have clarified all the concerns and provided strong evidence for the claims, and the work should now meet the high standard as expected by Nature Communications.

Minor Comments:

4. Resolution Calculation (Figure 5b2): The reported resolution value of 1.66 appears inconsistent with the lack of baseline separation in the chromatogram. The authors should verify their calculations.

Response: Thank you for this concern. The formula and calculation process for the resolution (R_s) are provided below. We have double-checked our calculations and confirmed that the value of 1.66 is correct. By definition, “baseline separation” is according to the widely accepted quantitative criterion of $R_s \geq 1.5$ (Colin F. Poole, *The Essence of Chromatography*, Section 1.6, 2002), which our result satisfies. We understand, however, a visual evaluation alone may not always accurately indicate whether peaks are baseline separated. To avoid any ambiguity, we have included a clear explanation of this definition in the revised manuscript.

$$R_s = \frac{2 \times (t_{R2} - t_{R1})}{1.7 \times (W_{1,h/2} + W_{2,h/2})} = \frac{2 \times (21.6 - 19.3)}{1.7 \times (0.70 + 0.93)} = 1.66$$

Meanwhile, we agree with your observation that Peaks (2+3) and Peak 1 did not appear to achieve complete baseline separation in the chromatogram (Figure 5b2). This occurred due to the overlapping between Peak 2 and Peak 3. With reference to Figure 5b3, Peak 2 eluted slightly earlier than Peak 3. As a result, the combined Peak (2+3) exhibited fronting rather than a Gaussian peak shape.

Changes to the Manuscript, Line 214: The core parameter for evaluating the separation performance of a critical pair is resolution (R_s), as defined by Equations 1-3. Baseline separation is quantitatively defined as $R_s \geq 1.5$.

5. Improved Data Representation: A plot of $\log(k)$ versus carbon number would better assess whether the separation follows a reversed-phase mechanism, as such a relationship should yield a linear trend.

Response: Thanks for this suggestion, Figure 4b has been replaced with a plot of $\log(k)$ versus carbon number. Updated Figure 4b shows an excellent linearity ($R^2 > 0.998$), confirming that the separation follows a reversed-phase mechanism.

Changes to the Manuscript, Figure 4:

Figure 4b. Comparison of retention factor of ordered mesoporous particle and totally porous particle.

Reviewer #2:

The manuscript by Prof. Zhang and coworkers reports the synthesis of monodisperse microparticles with a mesoporous nanostructure for chromatographic applications. The preparation of silica microparticles begins with the use of microfluidics to generate monodisperse droplets containing TEOS, hydrochloric acid, and an amphiphilic polymer surfactant. Subsequently, rapid solvent evaporation from the droplets induces self-assembly of the surfactant into a liquid crystalline phase, while simultaneously promoting the condensation of silica in the hydrophilic domains. Finally, hydrothermal treatment and calcination yield monodisperse silica particles with an ordered porous structure. By adjusting the SDAs and post-treatment conditions, the authors demonstrate that the nanostructures of the microparticles can be finely tuned. The authors further showcase the application of these particles in chromatographic separation, where they exhibit high performance compared to conventional silica particles.

We are grateful for your favorable and supportive evaluations. We also appreciate the detailed suggestions you have provided. In response to your comments, we have made the following revisions to improve the manuscript:

- (i) We have added high-magnification SEM images for all ordered mesoporous microspheres to better illustrate surface details.
- (ii) We have updated the comparison between ordered mesoporous particles and totally porous particles, with closely matched pore size, monodispersity, and particle size.
- (iii) We have introduced a comparative study of four different ordered mesoporous structures in the Supporting Information.
- (iv) We have also expanded the discussion on production yield and loading capacity of the ordered mesoporous material.

In my opinion, this work is very interesting; however, several issues still need to be addressed prior to publication:

1. Almost all of the SEM images lack surface details, making it difficult to extract useful information beyond particle size from the presented images.

Response: Thanks for this valuable suggestion. We have introduced high-magnification SEM images for the ordered mesostructured microspheres. This should enable our readers to have a clear insight of the surface details of these microspheres.

Changes to the Supporting Information:

Figure S5. SEM morphology characterization (a, c) with different magnifications and particle size distribution (b) of the cubic double gyroidal mesoporous silica microspheres templated by P123.

Figure S6. SEM morphology characterization (a, c) with different magnifications and particle size distribution (b) of the body centered cubic mesoporous silica microspheres templated by F127.

Figure S7. SEM morphology characterization (a, c) with different magnifications and particle size distribution (b) of the face centered cubic mesoporous silica microspheres templated by PEO₁₂₅-*b*-PMMA₂₄₉.

2. Why did the authors choose to use the particles mentioned in the main text (CV = 4.7%, Figure S14) for the separation performance comparison, rather than the 5 μm particles shown in Figure S12, which clearly have a more spherical morphology and more uniform size?

Response: Thank you for raising this concern. We selected the particles in Figure S14 (Figure S17 in revised Manuscript) for comparison based on their high separation performance, as a result of the high retentivity, despite their less spherical morphology and slightly higher CV value.

Due to a higher content of structure-directing agent used in the manufacturing, the particles in Figure S14 possess a larger pore volume and a higher orderliness of mesoporous structures, which inevitably resulting in a sphere-like polyhedron morphology composed of multiple planes. In our experiments, these properties are proved more conducive to achieving high efficiency and high retentivity, making the sphere-like polyhedral particles more suitable for separations of challenging critical pairs.

On the other hand, the actual difference in monodispersity between the two types of particles is not as large as that suggested by their CV values. Compared to the highly spherical particles, the sphere-like polyhedral particles cause greater error in particle size statistics, resulting in a larger apparent CV value. In theory, these two particles should exhibit similar monodispersity, as they both originate from monodispersed droplets (CV < 2%), which are generated from the same microfluidic chip.

3. In the comparison of separation performance, there are too many variables between the two types of particles, such as uniformity, pore structure, and morphology, making it difficult to draw accurate conclusions.

Response: We appreciate your insightful view. According to your suggestion, we have redesigned experiments for rigorous comparisons. Since commercially available packing particles with uniformity and pore structure identical to our ordered mesoporous particles (OMP) are lacking, we synthesized conventional totally porous particles for comparison, based on the method described in our previous work (*Angew. Chem. Int. Ed.*, 2025, 64, 202418642). The homemade totally porous particles (hTPP) exhibited morphological and pore structure parameters closely matching that of the OMP: in terms of morphology, hTPP and OMP showed comparable monodispersity (CV = 3.6% for hTPP and 4.7% for OMP) and a similar particle size of 5.0 μm; regarding pore structure, hTPP had a pore size of 8.2 nm (bare silica), which was reduced to 7.3 nm after C18 bonding, closely matching that of the OMP. It should be noted that the specific surface area of hTPP could not reach that of OMP, as the latter benefits from the

highly efficient utilization of pore space afforded by its ordered mesoporous architecture, enabling an extraordinary high specific surface area. Nonetheless, the hTPP still possess a specific surface area of 347 m²/g (bare silica), which, although lower than that of OMP, remains among the highest in comparison with commercial totally porous particles.

Upon controlling these parameters as closely as possible (as shown and listed in Figure S20 and Table S8), we further compared the chromatographic performance of the two types of particles. As shown in Figure S21 and Table S7, both efficiency and retentivity of hTPP are inferior to that of OMP, clearly demonstrating the meaningful improvements resulting from the ordered mesoporous structure.

Changes to the Supporting Information:

Figure S20. SEM morphology characterization (a) and particle size distribution (b) of the homemade totally porous silica microspheres.

Figure S21. Performance comparison between monodisperse ordered mesoporous particle and homemade totally porous particle. (a) Chromatograms of an alkylbenzene mixture on 15 cm long OMP and hTPP columns. Analytes: thiourea, methyl-, ethyl-, propyl-, and butylbenzenes (in order of elution); mobile phase: 60:40 v/v ACN/H₂O; UV detection: 214 nm. (b) Knox curves of OMP and hTPP.

Table S7. Properties of homemade totally porous particle.

Material	Silica	C18-Silica
Specific surface area (m ² /g)	347	208
Pore size (nm)	8.2	7.3

Pore volume (cm³/g)	0.78	0.55
Carbon load (%)	0	10.4
Particle size (μm)	5	5
Coefficient variance of particle size (%)	3.6	3.6
Ligand density (μmol/m²)	0	1.58

Table S8. Comparison of Knox coefficients and reduced plate heights obtained with TITAN-synthesized ordered mesoporous particles, commercial totally porous particles, and homemade totally porous particles.

	h_{\min}	a	b	c
OMP	2.78	0.65	4.83	0.14
TPP	4.08	1.05	5.14	0.32
hTPP	3.82	1.02	4.09	0.32

Changes to the Manuscript, Line 271: An additional set of totally porous particles (denoted as hTPP, for homemade totally porous particle) was intentionally synthesized to closely matched to OMP in terms of particle size, monodispersity, and pore size (Figure S20 and Table S8). As shown in Table S7 and Figure S21, OMP exhibited pronouncedly higher efficiency (2.78 vs. 3.82 for h_{\min}) and higher retentivity (19.2 vs. 4.0 for k) than that of hTPP, both outperforming the conventional totally porous counterparts.

4. The authors are encouraged to compare the influence of different nanostructures on separation performance to provide more comprehensive conclusions.

Response: Based on your suggestion, we have introduced performance comparison of the four different ordered mesoporous structures. Knox plots and separations of an alkylbenzene mixture were conducted, and the results have been added to the Supporting Information. These microspheres were controlled to have similar particle and pore sizes. The results show significant differences in efficiency and retentivity among the four ordered mesoporous structures.

Changes to the Supporting Information:

Figure S15. Performance comparison between microspheres with different ordered mesoporous structures, i.e., 2D hexagonal (2D-hex), cubic double gyroidal (cdg), body centered cubic (bcc) and face centered cubic (fcc) structures. (a) Chromatograms of an alkylbenzene mixture on 15 cm \times 100 μ m i.d. capillary columns. Analytes: thiourea, methyl-, ethyl-, propyl-, and butylbenzenes (in order of elution); mobile phase: 60:40 v/v ACN/H₂O; flow rate: 100 nL/min; UV detection: 214 nm. (b) Knox curves of microspheres with different ordered mesoporous structures.

Changes to the Manuscript, Line 230: As shown in Figure S15, the performance comparison of different ordered mesoporous structures demonstrated that, as expected, the 2D-hex structure indeed exhibited high retentivity and efficiency.

5. It would be helpful for the authors to clarify the production yield of such porous particles prepared by microfluidics, as well as the sample loading capacity that can currently be separated using their chromatographic system.

Response: Thanks for this suggestion, we have expanded discussions on both production yield and loading capacity. Particle yield depends on the droplet generation throughput. In this work, we employed a 120-channel array microfluidic chip based on our previous design (*Angew. Chem. Int. Ed.*, 2025, 64, 202418642), which generates droplets at an overall rate of \sim 240,000 per second with zero defective index, i.e. 100% production yield. This high-throughput process manufactures \sim 10⁹ microspheres per hour with a single chip, enabling the preparation of packing materials for one thousand 15 cm-long, 100 μ m i.d. chromatographic columns (\sim 10⁶ particles needed for one column), or ten 10 cm-long, 2.1 mm i.d. analytical scale columns (\sim 10⁸ particles needed per one).

Meanwhile, to determine the loading capacity of the ordered mesoporous particles, we used methylbenzene as a standard probe molecule and evaluated the column efficiency at different injection concentrations. The plot of column efficiency versus sample mass shows that the loading capacity—usually defined as the sample mass at which a 10% efficiency loss is observed (David V. McCalley, *J. Chromatogr. A*, 2008, 1193, 85-91)—was 0.087 μ g in this study.

Changes to the Manuscript, Line 406: The microfluidic platform enabled droplet generation at a super throughput of 240,000 Hz per chip, resulting in a production yield of approximately 10⁹ microspheres per hour.

Changes to the Manuscript, Line 257: The raised phase ratio also resulted in a high loading capacity of the OMP material, as demonstrated in the loading mass test (Figure S19).

Changes to the Supporting Information:

Figure S19. Column efficiency as the function of loaded mass for methylbenzene on 15 cm × 100 µm i.d. capillary columns. Mobile phase: 60:40 v/v ACN/H₂O; flow rate: 800 nL/min; UV detection: 214 nm. The efficiency loss was maintained within ~10% for solute loading mass up to 0.087 µg.

Reviewer #3:

The authors have presented a method for synthesizing mesoporous particles with a well-defined internal structure using a microfluidic method they refer to as TiTAN. This method is especially exciting for uses in chromatography, as the authors have demonstrated in their measurements showing superior separations of small molecules using particles with different internal structures and similar macroscopic (~4 µm) sizes. In my opinion, these are significant results in terms of the synthesis and application of the particles in an incredibly useful way -- and likely to be highly impactful. In fact, I can think of several more applications of these particles to fundamental scientific studies that extend beyond chromatography.

Nevertheless, as I am not an expert in chromatography, my review of this manuscript focuses exclusively on the authors' claims regarding the well-defined internal structure of the particles as probed using small-angle X-ray scattering (SAXS) and gas adsorption measurements. SAXS is one of the most appropriate techniques for measuring structure in systems such as this. Gas adsorption measurements similarly provide a good estimate of the pore size, and demonstrate the authors can tune this parameter with great precision. I have a few questions and concerns regarding these results, although overall I believe they strongly support the authors' conclusions.

We are deeply grateful for your positive and inspiring comments on our work. Your encouraging words are a great motivation for our research. We have read your suggestions carefully and made the following revisions to our work:

- (i) We have refined all SAXS figures to include labelling of peak positions relative to the fundamental peak (Q^*).

(ii) We have also expanded the discussion on pore size reproducibility, and introduced reproducibility data for pore size tuning between independent batches.

1. A comment: I would appreciate if the authors would label the relative peak positions (in terms of scattering vector Q) relative to the fundamental peak (Q^*) in their figures, rather than indexing the reflections. While this information is redundant, it would help the reader interpret these results. The authors should indicate each peak perhaps with an arrow of some sort. I realize this request may seem odd, but indexing the peaks this way makes it much easier for the reader to assess the quality of the mesopore structure. This is commonly done in the block copolymer community (see for example Chapter 5 in Basic X-ray Scattering for Soft Matter by de Jeu).

Response: Thank you for this excellent suggestion. We have revised all SAXS figures in the Manuscript in line with your suggestion. These revisions should enhance the clarity of our results to readers.

Changes to the Manuscript, Figure 1:

Figure 1. SAXS pattern (i) of the ordered mesoporous silica microspheres.

Changes to the Manuscript, Figure 2:

Figure 2. SAXS patterns of body centered cubic (c), face centered cubic (g), and cubic double gyroidal (h) configurations, respectively.

2. In some samples, for example the $Ia\bar{3}d$ /gyroid structures, the higher order reflections are quite weak (see Fig. 2g). How are the authors identifying peaks, and can they offer some insight into the relatively weak signals for higher values of Q ?

Response: Thank you for this insightful comment. In comparison with materials with $Ia\bar{3}d$ structure obtained through direct hydrothermal synthesis, those prepared via droplet evaporation followed by solvothermal treatment are prone to have asymmetric structural distortions during the evaporation process. Such distortions can reduce structural order and broaden diffraction peaks, which are further

amplified during solvothermal treatment, ultimately leading to poor long-range order and the absence of distinct reflections in the higher q region.

The structural assignment to $Ia\bar{3}d$ was made based on: (i) In comparison with untreated or hydrothermally treated samples that displayed a 2D hexagonal mesostructure, the SAXS patterns of the solvothermal treated samples exhibited clear differences in both peak positions and intensities, confirming the occurrence of a pronounced phase transition (Figure A2); (ii) When P123 was used as the structure-directing agent, the surfactants initially formed a 2D hexagonal ($p6mm$) mesophase (Figure A2b), upon solvothermal treatment in hexane at 80 °C, the hydrophilic PEO chains of P123 undergone partial dehydration and became hydrophobic, which reduced the V_H/V_L ratio (or increased the packing parameter, g), thereby driving the mesophase transition from 2D hexagonal to bi-continuous cubic $Ia\bar{3}d$ with lower interfacial curvature; and (iii) the TEM images no longer displayed 2D hexagonal channels but instead, revealed pore architectures consistent with $Ia\bar{3}d$ as reported in the literature (Facile Synthesis and Characterization of Novel Mesoporous and Mesorelief Oxides with Gyroidal Structures, *J. Am. Chem. Soc.*, **2004**, 126, 865-875) (Figure A3).

Taken together, these results strongly support our assignment of the $Ia\bar{3}d$ space group.

Figure A2: SAXS patterns of (a) hydrothermally treated, (b) direct calcined, and (c) solvothermally treated monodisperse microspheres prepared via TiTAN strategy.

Figure A3: TEM images of (a) hydrothermally treated, and (b) solvothermally treated monodisperse microspheres prepared via TiTAN strategy.

3. The authors show extraordinary control over the pore size (Fig. 3b). I assume these results are reproducible, but I wonder if the authors can comment on the variability between pore sizes between independent batches of particles? I may have missed this, but I cannot find in the manuscript or SI how many independent trials were performed. Were the nitrogen adsorption-desorption measurements

performed on independent batches? Is the high tunability of pore size that the authors demonstrated within a single batch of the particles? Etc. Some clarity regarding these points is necessary.

Response: We appreciate your comment regarding the pore size reproducibility. We have expanded the discussion on the variability in pore sizes between independent batches of particles. We intentionally collected droplets over a time span of three days and to produce three independent batches, respectively, with each batch divided into two groups subjected to different hydrothermal treatments. Nitrogen adsorption-desorption experiments were performed on independent batches to measure the pore size. As shown in Figure S8, for each batch, the two groups treated under distinct hydrothermal conditions exhibited angstrom-level differences in pore size, approximately 2-3 Å. Meanwhile, between different batches, the ordered mesoporous microspheres treated under the same hydrothermal conditions showed consistent pore size, with CV 2.1%, demonstrating good reproducibility.

Changes to the Supporting Information:

Figure S8. Reproducibility of pore size tuning via hydrothermal treatment. Droplets from three independent batches were collected over three days, and each batch was divided into two groups subjected to different hydrothermal treatments.

Changes to the Manuscript, Line 157: This angstrom-level pore size was found reproducible across three independent batches, with a CV of 2.1%, demonstrating good reproducibility of pore size tuning via hydrothermal treatment (Figure S8).

Other Revisions:

- (i) We have revised the format of the manuscript in accordance with *Nature Communications* formatting instructions.
- (ii) We have also noticed and corrected an editing error. The time unit labeled in Figure 4d should be “min” instead of “s”. This correction does not affect the conclusions, as the performance improvement of ordered mesoporous particles over totally porous particles remains unchanged.
- (iii) Three of the authors: Hanchen Cao, Jikai Chen, and Xiangyu Huang, since August 2025 will be continuously funded for their PhD studentship by the grant NSFC 22574134. These authors have contributed significantly to the revision of the current manuscript. In this regard, NSFC 22574134 is included in the Acknowledgement list.